# Data Consistent Deep Rigid MRI Motion Correction

**Nalini M. Singh**[1]                             NMSINGH@MIT.EDU
**Neel Dey**[1]                                   DEY@MIT.EDU
**Malte Hoffmann**[2,3]              MHOFFMANN@MGH.HARVARD.EDU
**Bruce Fischl**[1,2,3]                BFISCHL@MGH.HARVARD.EDU
**Elfar Adalsteinsson**[1]                    ELFAR@MIT.EDU
**Robert Frost**[*2,3]             SRFROST@MGH.HARVARD.EDU
**Adrian V. Dalca**[*1,2,3]                    ADALCA@MIT.EDU
**Polina Golland**[*1]                POLINA@CSAIL.MIT.EDU
[1] *Massachusetts Institute of Technology*
[2] *Athinoula A. Martinos Center for Biomedical Imaging*
[3] *Harvard Medical School*
[*] *Equal Contribution*

**Editors:** Accepted for publication at MIDL 2023

## Abstract

Motion artifacts are a pervasive problem in MRI, leading to misdiagnosis or mischaracterization in population-level imaging studies. Current retrospective rigid intra-slice motion correction techniques jointly optimize estimates of the image and the motion parameters. In this paper, we use a deep network to reduce the joint image-motion parameter search to a search over rigid motion parameters alone. Our network produces a reconstruction as a function of two inputs: corrupted k-space data and motion parameters. We train the network using simulated, motion-corrupted k-space data generated with known motion parameters. At test-time, we estimate unknown motion parameters by minimizing a data consistency loss between the motion parameters, the network-based image reconstruction given those parameters, and the acquired measurements. Intra-slice motion correction experiments on simulated and realistic 2D fast spin echo brain MRI achieve high reconstruction fidelity while providing the benefits of explicit data consistency optimization. Our code is publicly available at https://www.github.com/nalinimsingh/neuroMoCo.

## 1. Introduction

Subject motion frequently corrupts brain magnetic resonance imaging (MRI) and obfuscates anatomical interpretation (Andre et al., 2015). *Prospective* motion correction strategies adapt the acquisition in real-time to adjust for measured rigid-body motion (Tisdall et al., 2012; Frost et al., 2019). Unfortunately, prospective strategies require altering clinical workflows, prolonging scantime, or interfering with standard acquisition parameters. *Retrospective* strategies correct motion algorithmically after acquisition, with or without additional motion measurements (Batchelor et al., 2005; Bammer et al., 2007; Polak et al., 2022). Retrospective motion correction without additional motion information is particularly appealing, because it does not require external hardware or pulse sequence modifications and enables retroactive correction of large, previously collected k-space datasets. This

work develops a deep learning method for retrospective rigid motion correction. We focus on *intra-slice* motion correction in multi-shot acquisitions, which acquire multiple k-space segments per slice and are a workhorse of clinical screening exams (Mehan et al., 2014).

Retrospective intra-slice motion correction is often formulated as joint optimization of rigid motion parameters and the underlying image (Haskell et al., 2019; Cordero-Grande et al., 2016). The result is a highly non-convex, ill-posed optimization problem with undesirable local minima. The challenging nature of joint estimation has inspired several deep learning alternatives that train standard supervised convolutional neural networks (CNNs) to reconstruct motion-free images directly from motion-corrupted inputs (Pawar et al., 2018; Duffy et al., 2021; Levac et al., 2022b; Singh et al., 2022). Alternatively, GANs can be used to provide priors on the reconstructed images (Küstner et al., 2019; Johnson and Drangova, 2019). None of these methods incorporate the MR forward model or allow the reconstruction strategy to depend on the motion parameters. Further, while these approaches provide visually appealing reconstructions, they do not enforce *data consistency* between the estimated output and the acquired measurements. As a result, they are susceptible to producing hallucinations, i.e., visually plausible features inconsistent with the acquired measurements.

Our work is closely related to methods that combine neural networks with enforced data consistency. Haskell et al. (2019) use an iterative procedure where a CNN produces an initial motion-corrected image, which initializes an alternating optimization over the motion parameters and underlying image. Levac et al. (2022a) use a score-based generative model that estimates the log probability of a reconstruction. At test-time, motion parameters and the underlying image are optimized to find a solution that is both data consistent and has a high prior probability under the generative model. Both methods incorporate optimizations of the underlying image and motion parameters. Our proposed strategy reduces this joint optimization to an optimization over just the motion parameters.

Our key insight is that the joint optimization can be simplified using a deep neural network to learn a mapping from proposed motion parameters and corrupted k-space to a motion-corrected reconstruction. Then, inference simplifies to a test-time search solely over the motion parameters, which in turn imply a reconstructed image. This strategy eliminates the search over images while retaining the benefits of iterative, optimization-based motion parameter estimation. In particular, the discrepancy between acquired measurements and the estimated image and motion parameters can be automatically monitored during the optimization to reject failure cases with a poor quality reconstruction.

We demonstrate our intra-slice motion correction method on 2D fast spin echo (FSE) brain MRI. Our method produces reconstructions that are consistent with the acquired k-space measurements in the presence of inter-shot motion or provides an indicator when they are not. The method is trained on simulated pairs of corrupted and motion-free examples and generalizes to a proof-of-concept acquired k-space example. In both simulated and realistic data, our method achieves consistently high quality reconstruction and motion parameter estimation consistent with the forward imaging model.

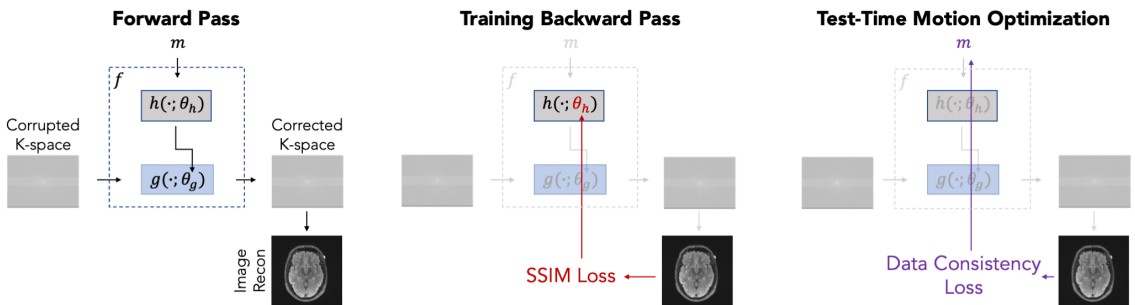

**Fig. 1:** Information flow through our network. During a forward pass, true or estimated motion parameters $m$ serve as input to hypernetwork $h(\cdot; \theta_h)$ which generates the weights $\theta_g$ of a reconstruction subnetwork $g(\cdot; \theta_g)$. Reconstruction subnetwork $g(\cdot; \theta_g)$ takes corrupted k-space data as input and produces a reconstruction. The hypernetwork weights $\theta_h$ are the only network parameters directly updated during training. At test-time, we freeze $\theta_h$ and use the data consistency loss to optimize the motion parameters $m$.

## 2. Methods

There are two stages to our motion correction procedure: (1) training a network that maps motion parameters to reconstructions and (2) performing a test-time optimization that solves for the motion parameters for an unseen k-space example, which then maps to the final reconstruction via the trained network (Fig. 1). We first describe the MRI forward model and then detail its use in neural network training and motion parameter estimation.

**Forward model.** MRI acquires Fourier measurements described by the forward model

$$y_i = A_i x + \epsilon, \tag{1}$$

where $x$ is the underlying 2D image we wish to recover, $y_i$ are the acquired MRI measurements from the $i^{th}$ coil element, and $\epsilon$ is i.i.d. complex-valued Gaussian noise. In the ideal setting where no motion is present, the forward imaging operator $A_i$ for the $i^{th}$ coil is

$$A_i = UFC_i, \tag{2}$$

where $C_i$ denotes a diagonal matrix that encodes the coil sensitivity profile of the $i^{th}$ coil, $F$ is the 2D Fourier transform, and $U$ encodes the undersampling pattern for the acquisition.

In the presence of motion, the forward operator $A_i$ is a function of unknown motion parameters $m$. We consider in-plane rigid-body motion and assume a quasi-static motion model where the 2D object moves between shots but remains stationary during the acquisition of each individual shot. Under this model,

$$A_i(m) = \sum_s U_s FC_i M_s(m), \tag{3}$$

where $U_s$ encodes the undersampling pattern for shot $s$ and $M_s(m)$ is a motion matrix encoding in-plane rigid translation and rotation for shot $s$. This forward model differs from Eq. 2 in two ways: (1) it varies across acquired images, which are characterized by different motion parameters, and (2) it is in general unknown because $m$ is unknown.

**Training.** We use the forward model in Eqns. 1 and 3 to simulate data to train a network $f(y, m; \theta)$ that takes corrupted k-space data and motion parameters as input. The parameters $\theta$ are trained using simulated, motion corrupted data:

$$\theta^* = \arg\min_{\theta} \mathbb{E}_{x,m} L(f(A(m)x + \epsilon, m; \theta), x), \tag{4}$$

where $L$ is the image reconstruction loss. In this paper, we use the negative structural similarity index measure (SSIM) (Wang et al., 2004) loss, but our method accepts any differentiable reconstruction loss function.

**Test-time optimization.** Assuming Gaussian noise in Eq. 1, Bayes' rule yields

$$\begin{aligned} \log p(x, m|y) &= \log p(y|x, m) + \log p(x) + \log p(m) + c \\ &= -\frac{1}{2\sigma^2}||y - A(m)x||^2 + \log p(x) + \log p(m) + c, \end{aligned} \tag{5}$$

where we have assumed the underlying image $x$ and motion parameters $m$ to be independent. All constants have been consolidated into $c$ and $\sigma$ is the standard deviation of the additive Gaussian noise. Maximizing the posterior probability $p(x, m|y)$ involves minimizing the *data consistency* loss $||y - A(m)x||^2$ and maximizing priors over the underlying image and motion parameters. The data consistency loss measures disagreement between the acquired measurements $y$ and predicted signals $A(m)x$, while the priors are often expressed as handcrafted regularizers (Rudin et al., 1992) or explicitly or implicitly modeled by a neural network (Goodfellow et al., 2020; Kingma and Welling, 2013; Song et al., 2020).

At test-time, we freeze the weights of the reconstruction network and optimize motion parameters with a data consistency loss for acquired (corrupted) measurements $y$:

$$\hat{m}(y) = \arg\min_{m} ||y - A(m)f(y, m; \theta^*)||^2. \tag{6}$$

The final image reconstruction is then simply $\hat{x}(y, \hat{m}) = f(y, \hat{m}; \theta^*)$. Monitoring the loss in Eq. 6 provides information about how well the motion correction procedure is performing. When this loss is high relative to the total spectral energy of the acquired k-space data even after optimization, we discard the reconstruction and/or decide to reacquire the image.

## 3. Network Architecture

Unlike standard supervised deep learning methods which do not incorporate information about the motion parameters into the structure of the neural network (Pawar et al., 2018; Duffy et al., 2021; Levac et al., 2022b; Singh et al., 2022; Küstner et al., 2019; Johnson and Drangova, 2019), our method uses a motion-dependent neural network. In this paper, we instantiate the reconstruction network $f(y, m; \theta)$ with a hypernetwork $h(\cdot; \theta_h)$ (Ha et al., 2016; Hoopes et al., 2021) operating on motion parameters $m$ to generate weights $\theta_g$ of a reconstruction subnetwork $g$:

$$f(y, m; \theta) = g(y; \theta_g(m)) \text{ where } \theta_g(m) = h(m; \theta_h). \tag{7}$$

The weights $\theta_g$ of the reconstruction subnetwork are never trained directly. The only trainable parameters in $f$ are $\theta_h$, the weights of the network $h$ that produces $\theta_g$. This hypernetwork architecture enables flexible prediction of a reconstruction subnetwork $g(\cdot; h(m; \theta_h))$ specific to the motion parameters, just as $A(m)$ is a function of the motion parameters.

The inputs to our hypernetwork $h$ are the motion parameters $m$ comprising 18 scalars representing x- and y- translations and a rotation for each of 6 k-space shots. $h$ consists of 6 fully connected layers with 256 units followed by convolutions to produce the reconstruction subnetwork weights $\theta_g$; $h$ contains 2.7M trainable parameters.

The input to the reconstruction subnetwork is the Autocalibrating Reconstruction for Cartesian imaging (ARC) reconstruction (Brau et al., 2008) of the acquired signals, expressed as 88-channel real and imaginary components of k-space data from 44 receive coils (4 neck channels were discarded). This input is processed via 6 convolution layers in both image and frequency space (Singh et al., 2022), where each $3 \times 3$ convolution outputs 32 channel features. The reconstruction subnetwork $g$ outputs 88 k-space channels. We reconstruct the final image via the inverse Fourier transform and root-sum-of-squares coil combination.

We note that several other architectures are possible for $f(y, m; \theta)$, e.g., an architecture that takes motion parameters as additional network inputs. Identifying the optimal architecture for motion-parameterized neural networks is an important avenue for future research. Here, we use hypernetworks as one example formulation of $f(y, m; \theta)$ to demonstrate our general approach of test-time motion parameter estimation to produce a reconstructed image consistent with the acquired data.

## 4. Experiments

**Data.**  We demonstrate our approach on 2D T2 FLAIR FSE brain MRI k-space data (3T GE Signa Premier, 48-channel head coil, 6-shot acquisition, TR=10s, TE=118ms, TI=2.6s, FOV=$260 \times 260$mm$^2$, slice thickness=5mm, slice spacing=1mm, acceleration factor R=3) under an approved IRB protocol. The shots are designed such that the second shot contains the central k-space line and the highest spectral energy. The first, third, and fourth shots contain comparable spectral energy, while the fifth and sixth shots largely contain data from the periphery of k-space and have on the order of one-thousandth the spectral energy in the second shot. We split the dataset into 553/197/100 training/validation/test 2D slices from 31/11/9 subjects respectively, with no subject overlap. We treat the acquired data as motion-free ground truth and simulate motion artifacts during training and testing. We also generate a realistic test example by mixing acquired k-space measurements $y_{i,1}$ and $y_{i,2}$ from two scans of the same subject in different head positions: $y_i = U_{pre}y_{i,1} + U_{post}y_{i,2}$.

**Motion simulation.**  We simulate motion-corrupted data from a collection of acquired motion-free k-space data. We apply ARC reconstruction (Brau et al., 2008) to the data followed by the inverse Fourier transform and root-sum-of-squares coil combination, yielding motion-free image $x$. We estimate coil sensitivity profiles $S_i$ using ESPIRiT (Uecker et al., 2014; Iyer et al., 2020) and extend the profiles to the image edge via B-spline interpolation. We synthesize motion-corrupted measurements using Eqns. 1 and 3. We define motion matrix $M$ by selecting a random shot affected by motion and sampling 2D translation parameters $(\Delta_h, \Delta_v) \sim \mathcal{U}(-10\text{mm}^2, 10\text{mm}^2)$ and rotation parameter $\theta \sim \mathcal{U}(-10°, 10°)$. We apply the sampled motion parameters to the shots including and following the randomly selected motion-affected shot. We add complex-valued Gaussian noise ($\sigma = 10,000$) to the simulated k-space as in Eq. 1 such that the noise comprises on average 5% of each coil's spectral energy. The motion simulation is detailed in Appendix A, Fig. 5.

**Implementation details.**   We normalize input/output k-space by the maximum intensity in the corrupted image. Networks are trained to reconstruct the image ($x$ or $M(m)x$) corresponding to the central k-space line. All models are trained with the SSIM loss function using the Adam optimizer (learning rate $10^{-3}$) and batch size 6. The test-time optimization uses 4 trials of gradient descent. In experiments with simulated data, we use a cyclical exponential decay schedule from $10^{-6}$ to $10^{-7}$, while on the realistic test example we fix the learning rate at $10^{-9}$. We tune hyperparameters and perform model selection on the validation set and hold out the test set for final evaluation. Each network is trained for 250,000 iterations. Optimizing the motion parameters takes $\sim$5 minutes per 2D slice to run the 4 trials sequentially and is trivially parallelizable. After reconstruction, we automatically reject poor reconstructions indicated by a data consistency loss greater than 5% of the total spectral energy of the input k-space measurements.

**Baselines.**   We analyze two versions of our method: `HN` (Hypernetwork), which includes all results of our test-time optimization, and `HN-R`, which rejects examples where the data consistency loss is high relative to the spectral energy. We compare against four baselines.

   `ARC` (Brau et al., 2008) is a classical autocalibrating parallel imaging reconstruction that interpolates undersampled k-space regions based on a fully-sampled central calibration region. This method performs no motion correction and is commonly used clinically.

   `Conv` (Singh et al., 2022) uses $g(\cdot; \theta_g)$ with $\theta_g$ trained directly. The network has no dependence on motion parameters and requires no test-time optimization. The kernel size and number of features are varied to match the number of trainable parameters in `HN`. This baseline characterizes a state-of-the-art deep learning alternative to `ARC` and is an ablation that isolates the difference between a standard network and our hypernetwork.

   `Model-Based-GT` (Gallichan et al., 2016) assumes known motion parameters and applies analytical corrections. Translations and rotations are corrected via k-space phase shifts and rotations and the result is reconstructed via an inverse Non-Uniform Fast Fourier Transform (NUFFT) (Fessler and Sutton, 2003; Muckley et al., 2020). This method is an upper bound on joint estimation performance because it uses ground truth motion parameters.

   `HN-GT` computes the output of our hypernetwork when ground truth motion parameters are provided as input. This characterizes a motion-informed deep learning method and, similar to `Model-Based-GT`, represents the best-informed scenario for reconstruction.

## 5. Results

Fig. 2 shows reconstructions on a simulated example. Our method `HN` produces sharper, more accurate reconstructions than 'motion-naive' `ARC` and `Conv` that do not incorporate motion information. It also provides reconstructions of similar quality to 'motion-aware' `Model-Based-GT` and `HN-GT` that have access to ground truth motion parameters.

   Fig. 3a shows that our method outperforms motion-naive methods across the test set and is on par with the motion-aware methods. This result holds when other image quality metrics are used (Appendix B, Fig. 6). Fig. 3b shows that our method yields better reconstructions than motion-naive baselines for the vast majority of subjects. Fig. 3c demonstrates that our method recovers accurate motion parameters for the first four high-energy shots, with most inaccurate cases automatically rejected.

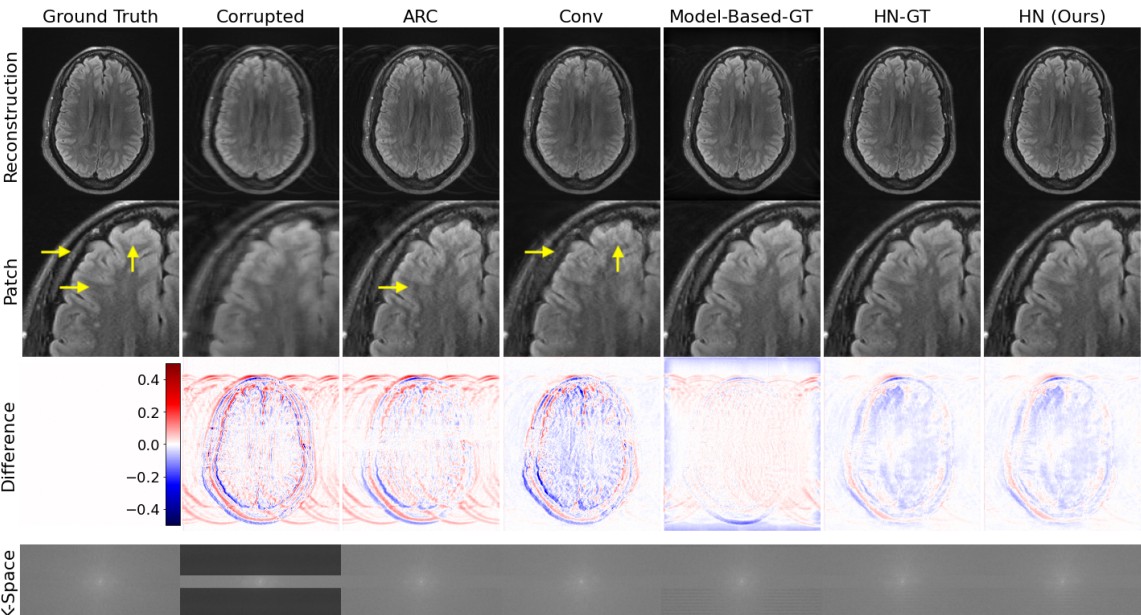

**Fig. 2:** Simulated test example. `HN` outperforms motion-naive baselines (`ARC` and `Conv`) and performs similarly to motion-aware methods (`Model-Based-GT` and `HN-GT`) without access to any motion parameters. Yellow arrows highlight local reconstruction errors.

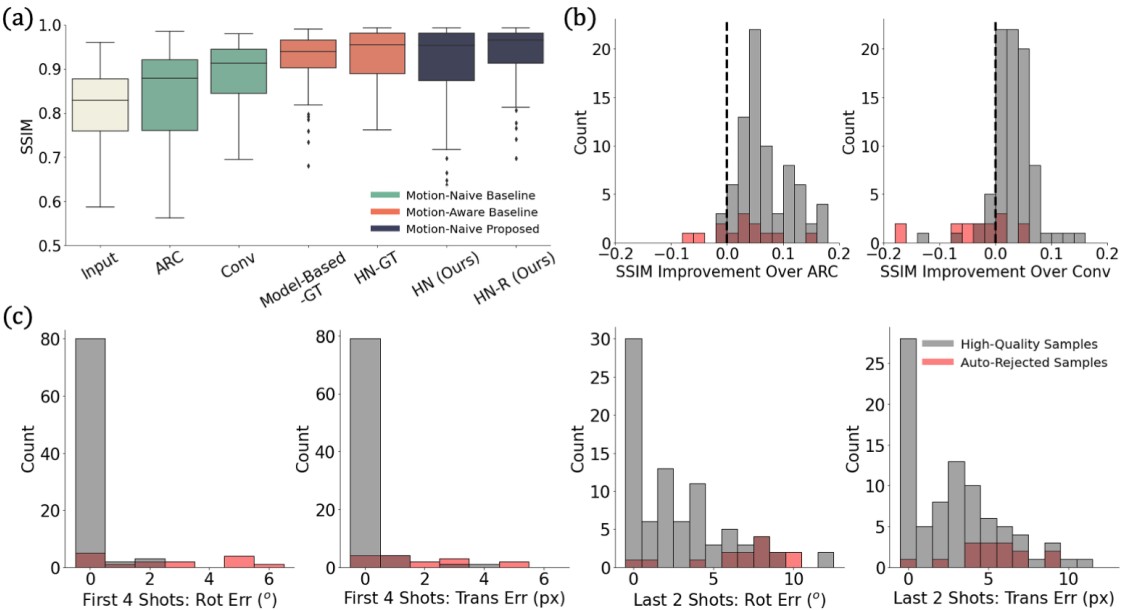

**Fig. 3:** Simulation results. (a) Reconstruction SSIM across methods. (b) SSIM improvement over motion-naive methods. We improve on the baselines for bars right of the dashed line. (c) Motion estimate errors from our method in the four high-energy and two low-energy shots. Our method outperforms motion-naive methods and is on par with motion-aware ones. Our automated rejection strategy effectively identifies optimization failures.

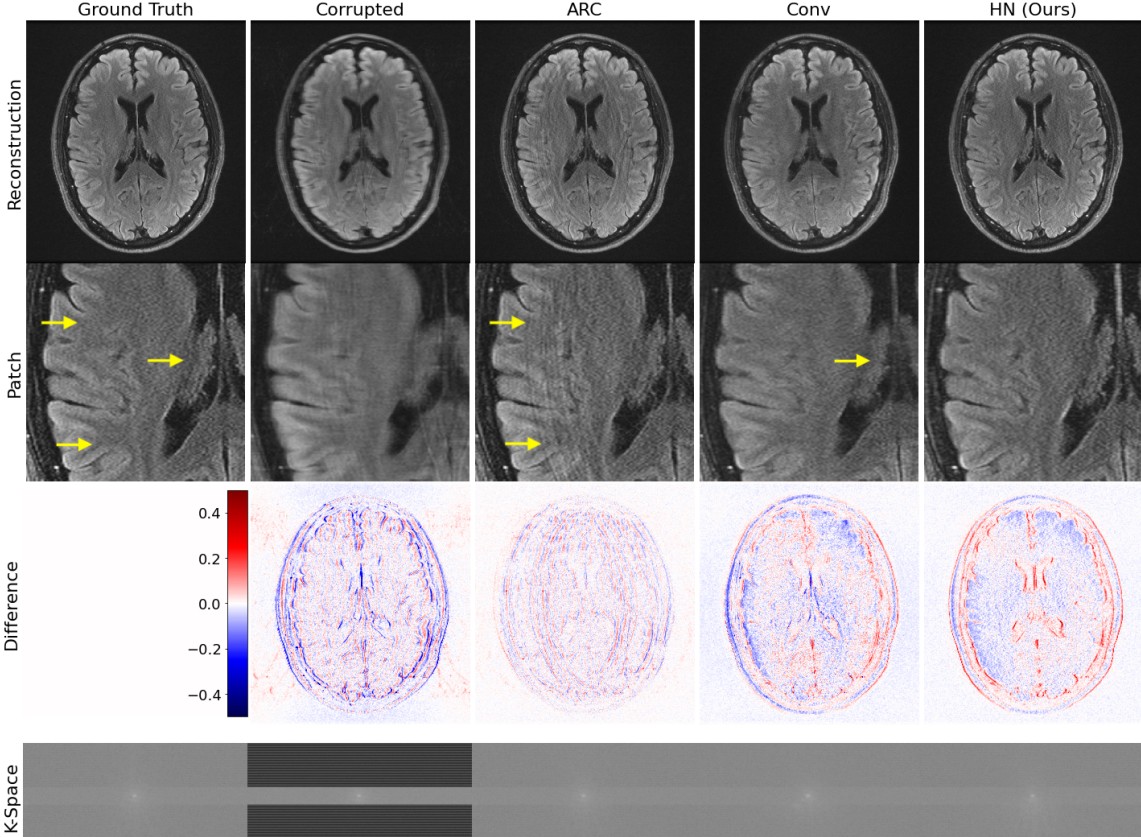

**Fig. 4:** Realistic test example. For this example, we do not have access to the true motion parameters, and our method (`HN`) outperforms the baselines which do not require this information. `HN` removes the artifact in `ARC` and is sharper than `Conv` (see arrows). Despite being trained on simulated data, our method generalizes to this test example based on acquired k-space data and produces a high-quality reconstruction.

Our method does not reliably recover motion parameters for the last two shots. These low-energy shots play a relatively small role in the data consistency optimization but also have low impact on the resulting reconstruction. Of fifteen automatically rejected cases, motion occurred between the second and third shots in nine examples. This splits the first four high-energy shots between two positions and causes hard-to-correct artifacts.

Finally, Fig. 4 visually illustrates reconstruction for the realistic test example that combines acquired k-space data from the same subject in two different positions. `HN` provides the best reconstruction among methods that do not require access to the true motion parameters, demonstrating that it successfully handles real motion-corrupted signals despite using only simulated data during training.

## 6. Discussion and Conclusions

This work develops a general deep rigid motion correction approach for multi-shot MRI that we evaluate on 2D FSE T2 FLAIR data. The proposed framework first learns a mapping between corrupted k-space data, true motion parameters, and high-quality reconstructions. At test-time, only the motion parameter estimates are optimized, yielding data-consistent reconstructions, thus alleviating the complexity of a joint search over images and motion parameters. The approach includes a strategy for automatically rejecting samples where the optimization fails. Our method reliably produces high-quality reconstructions in simulation and generalizes to a proof-of-concept realistic acquired k-space example.

While we demonstrate that our approach generalizes to acquired k-space data, our evaluation focuses on simulated data. We do not model through-plane motion, a known source of significant artifact. Less well-studied effects including intra-shot motion, spin history, and signal decay during the FSE echo train may also affect our method.

Future work will thus investigate, model, and correct these additional motion effects while retaining data-consistent reconstructions in larger scale, clinical k-space datasets. Beyond MRI, our strategy for learning physically consistent reconstructions given a partially-unknown forward model could be applied to other semi-blind inverse problems, e.g. in seismic imaging or computational photography.

## Acknowledgments

Research reported in this paper was supported in part by GE Healthcare and by computational hardware provided by the Massachusetts Life Sciences Center. We also thank Steve Cauley for helpful discussions. Additional support was provided by NIH NIBIB (5T32EB1680, P41EB015896, 1R01EB023281, R01EB006758, R21EB018907, R01EB019956, R01EB017337, P41EB03000, R21EB029641, 1R01EB032708), NIBIB NAC (P41EB015902), NICHD (U01HD087211, K99 HD101553), NIA (1R56AG064027, 1R01AG064027, 5R01AG008122, R01AG016495, 1R01AG070988, RF1AG068261), NIMH (R01 MH123195, R01 MH121885, 1RF1MH123195), NINDS (R01NS0525851, R21NS072652, R01NS070963, R01NS083534, 5U01NS086625, 5U24NS10059103, R01NS105820), the Blueprint for Neuroscience Research (5U01-MH093765), part of the multi-institutional Human Connectome Project, the BRAIN Initiative Cell Census Network grant U01MH117023, and a Google PhD Fellowship.

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

## Appendix A. Motion Simulation

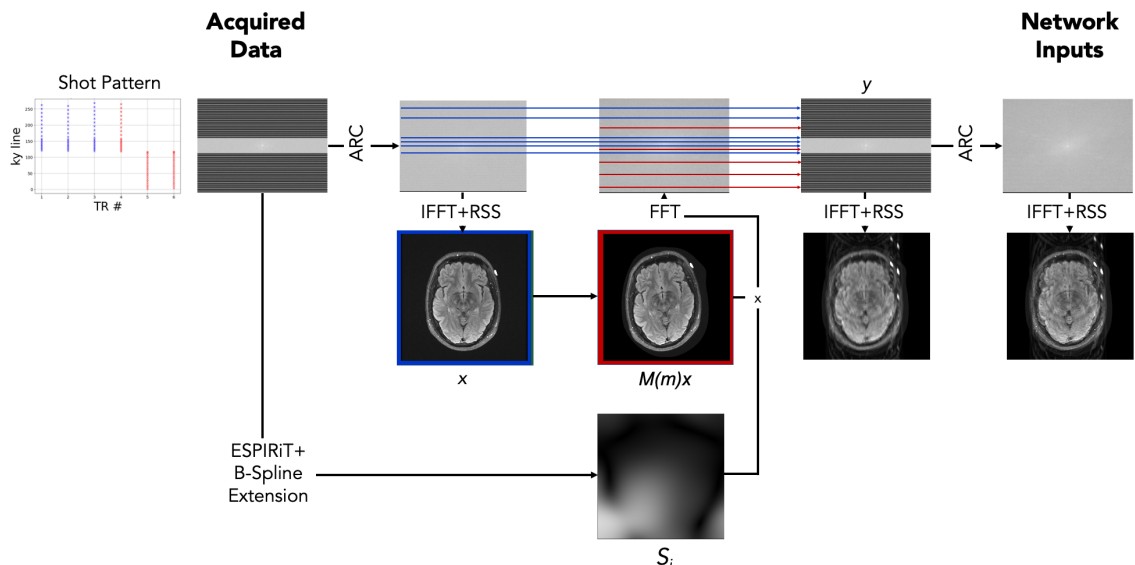

**Fig. 5:** Motion simulation schematic. We apply ARC reconstruction (Brau et al., 2008) to motion-free data followed by the inverse Fourier transform and root-sum-of-squares coil combination, yielding the initial motion-free image $x$. We also estimate coil sensitivity profiles $S_i$ from the acquired data using ESPIRiT (Uecker et al., 2014) with learned parameter estimation (Iyer et al., 2020) and extend the profiles to the image edge via B-spline interpolation. Next, we simulate an image under a sampled random motion $M(m)$ by applying rotations and translations to the image and use the sensitivity maps and a Fourier transform to simulate k-space data corresponding to the moved position. Based on the shot pattern for the acquisition, we combine k-space data from the appropriate lines corresponding to the position pre- (blue) and post-motion (red) to form simulated, motion-corrupted k-space measurements $y$. This simulates the k-space data that would have been acquired had the subject moved from the blue position to the red position over the course of the acquisition. The input to our networks is the ARC reconstruction of this simulated $y$. In practice, we sample two versions of $M(m)x$ and treat one of the two as $x$, to avoid discrepancies between simulated and acquired data when mixing the k-space.

## Appendix B.  Additional Metrics

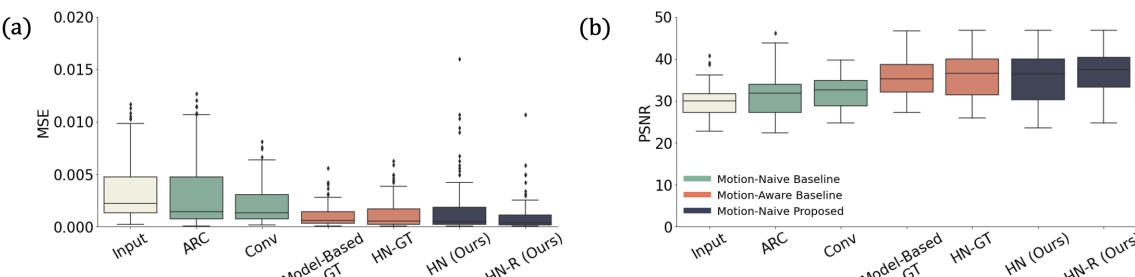

**Fig. 6:** Reconstruction quality of all methods as measured by mean square error (MSE) and peak signal-to-noise ratio (PSNR). As with SSIM (Fig. 3a), our method outperforms motion-naive baselines and performs comparably to motion-aware ones according to these metrics.

