# OpenReview forum: "Data Consistent Deep Rigid MRI Motion Correction"
_MIDL.io/2023/Conference — MIDL 2023 Oral_

### Official Review · Reviewer_tDsZ · 2023-02-02

**Confidence:** 4
**Preliminary Rating:** 4

**Summary:**

This paper describes a new method for dealing with motion artifacts in MRI.  The approach uses two components: an image reconstruction network that performs reconstruction based on known motion parameters, together with a search over motion parameters.  Validation is performed using simulations which confirm improve performance.

**Strengths:**

The ideas in the paper are new and interesting.  The paper also presents MRI physics accurately, and does an excellent job of explaining its own limitations.  This is very important, but is unfortunately becoming less and less common in the MRI machine-learning literature.

**Weaknesses:**

The noise model in Equation 1 is wrong.  In k-space, the noise should be complex Gaussian, not Rician.  The Rician distribution would only arise if the phase information were thrown away, which would never be done in k-space.

The reconstruction subnetwork theta_g is never explained, but seems essential for this approach to be plausible.

It's not clear how much motion is being simulated, which makes it difficult to have a real appreciation of the results.

**Deanonymize Review:**

no

**Paper Type:**

methodological development

**Questions To Address In The Rebuttal:**

It would be good for the authors to address the comments listed under Weaknesses, but I don't have other questions for the authors to address in their rebuttal.  Nothing else to say (but need to fill the character quota).

---

### Official Review · Reviewer_iAch · 2023-02-03

**Confidence:** 4
**Preliminary Rating:** 5
**Recommendation:** Oral, Poster

**Summary:**

The manuscript "Data Consistent Deep Rigid MRI Motion Correction" presents a timely approach to tackle rigid motion in magnetic resonance imaging (MRI), which is one major problem in various imaging settings. The suggested method splits the problem in two sub-tasks to be optimized nearly independently. A hypernetwork is optimized to provide (normally unknown) motion parameters. This hypernetwork predicts weights that are used by a second subnetwork, called reconstruction network, to predict motion corrected k-space data which are then processed in a standard fashion (Fourer transform + coil combine). While training was done on data with simulated motion, the performance of the network was also tested on a real-world data.
The introduction motivates the work and gives a short overview of the topic including other relevant methods. The methods section is clearly structured and guides the reader well. In 3. Experiments the relevant information for the experimental setup are described and these allow the reader to learn about the most interesting aspects, such as used data, data preparation, network architecture and baseline / reference methods. The results are well presented and the manuscript ends with a fair discussion and conclusions section in which also the limitations and future work ideas are reported.
In summary, while the question / task (use data-driven method to retrospectivly correct motion-defected MRI data) is not new, the presented method with two subnetworks / optimizations is new and appealing.


**Strengths:**

Overall, the presented results are convincing also other, existing methods show comparable performance (Model-Based-GT), however the key aspect in the presented work is it's good performance for the cases where motion parameters are unknown (this is often the case or additional effort/measurements are required to track motion). The experimental setup is good and well thought through. Good practise was applied in data organisation.
The results are briefly but clearly discussed and limitations are named. Furthermore, the work showed the performance of the approach on real-world data, which is of course highly relevent, even though the real-word experiment might still not reflect 100% the final szenario (more severe/complex motion).
The comparison to alternative methods and baseline is fair and comprehensive. Ranging form (clinical) standard, to analytically and data-driven state-of-the-art methods.


**Weaknesses:**

Only SSIM was considered. Even tough the SSIM is kind of well established, it has also become apparent that one metric alone is often not enough to fully quantify image quality. MSE oder PSNR are just to other metrices that would have been interesting and are also often reported along SSIM.

**Deanonymize Review:**

yes

**Detailed Comments:**

- 1. Introduction: "Retrospective motion correction without additional ... collected k-space datasets". True statement, but necessarily not the optimal way (alone). Also with DL-solutions it probably remain true, that any method relies on its data (Rubish in rubish out). Totally motion-defected data can not be recreated with full fidalety. In particular, for through-plane motion retrospective approaches have limited remedy.
- 3. Experiments/Motion simulation: ARC not introduced (Autocalibrating Reconstruction for Cartisian imaging) Have you used parallel imaging in your 2D T2 FLAIR FSE sequence? If so, please specify the acceleration in the section above (3. Experiments/Data).
- 3. Experiments/Implementation details: HN not introduced (hypernetwork).
-Fig 2 & 4: LUT / colormap for Difference subimages would be helpful. What do the yellow arrows indicate. Some details about what is indicated by these would be helpful in the caption.
3. Experiments/Baselines - last sentence: "...respresents the best-case scenario for reconstruction". I understand what you want to say, but best-case sounds a little like results/discussion, for methods section I would choose more careful wording: "...represents the best-informed scenario..."

**Paper Type:**

methodological development

**Questions To Address In The Rebuttal:**

Overall I can strongly accept the submitted manuscript. In particular, the addition of at least a second metric for image quality quantification would make the work more convincing.
Furthermore, the discussion of the extension of this method to other sequences (e.g. 3D, FLASH, other trajectories) would be very interesting.

---

### Official Review · Reviewer_XDx7 · 2023-02-06

**Confidence:** 4
**Preliminary Rating:** 5
**Recommendation:** Oral

**Summary:**

This paper proposes a novel method for data-consistent multi-shot MRI 2D motion correction and image reconstruction. The novelty of the approach is learning a motion conditioned hypernetwork that produces the parameters for a multi-channel MRI reconstruction model that is aware of subject motion. Following training, this model can be used at test time by optimising the motion parameters to improve the consistency with the input k-space data. This approach is demonstrated on simulated motion in fast spin echo brain MRI.

**Strengths:**

The core idea of this paper is an interesting one, conditioning the MRI reconstruction function on the predicted motion. This leads to an effective approach to maintain consistency with the original k-space data and allows for good estimation of motion and rejection of poor estimates. The paper is well explained and the model description is generally clear and well argued. It's good the see the authors worked with the original multi-channel k-space data. The results section covers simulated experiments, which seem to have been conducted fairly and the quantitative and qualitative analysis has a good level of detail.

**Weaknesses:**

The formulation as a motion dependent hyper-network seems reasonable, although it's not clear why this is preferable to incorporating the motion parameters as input to the reconstruction model $g$. I don't believe this baseline was included in the comparison.

Eq 1. contains the forward model for k-space where $\epsilon$ is defined as Rician noise, but this would only be for magnitude k-space data and you have complex k-space data, correct? If your formulation was $A_i(x+\epsilon)$ and $x$ is a magnitude image then that would also make sense. Moreover, in equation 6 your data consistency term is squared error (equivalent to a Gaussian) rather than Rician so inconsistent with this statement.

A couple of details, e.g. why SSIM is used in eq. 5 rather than a log likelihood? and some idea of what $\epsilon$ in eq. 5 refers to, is it sampled noise?

As was noted honestly by the authors, the 2D only motion estimation and reconstruction is a practical deficiency, as is the reconstruction time of 5 minutes per slice.

Minor:
Eq. 6 uses an L2 norm - but describes it as a probability. Why not state it as a multivariate normal log prob for consistency?

**Deanonymize Review:**

no

**Paper Type:**

methodological development

**Questions To Address In The Rebuttal:**

I think this paper is of sufficient interest and quality to be presented at MIDL. Although the work clearly has some limitations, these are honestly addressed by the authors. An added comparison with a non-hypernetwork based model would be good to include, as well as addressing the minor issues highlighted in the weaknesses.

---

### Meta-Review · Area_Chair_DYWn · 2023-02-24

**Recommendation:** Accept (Oral)
**Confidence:** 4

**Metareview:**

The reviewers unanimously agree that the approach proposed is novel and tackles a timely problem, that the experiments are well performed and the results convincing. They appreciated the honesty of the authors regarding the limitations of their work and found the manuscript clear.